# Dimensional Stability of Mirror Substrates Made of Silicon Particle Reinforced Aluminum

**DOI:** 10.3390/ma15092998

**Published:** 2022-04-20

**Authors:** Jan Kinast, Andreas Tünnermann, Andreas Undisz

**Affiliations:** 1Fraunhofer Institute for Applied Optics and Precision Engineering IOF, Albert-Einstein-Str. 7, 07745 Jena, Germany; andreas.tuennermann@uni-jena.de; 2Institute of Applied Physics, Abbe Center of Photonics, Friedrich Schiller University, Max-Wien-Platz 1, 07743 Jena, Germany; 3Institute of Materials Science and Engineering, Chemnitz University of Technology, Erfenschlager Str. 73, 09125 Chemnitz, Germany; andreas.undisz@mb.tu-chemnitz.de; 4Otto Schott Institute of Materials Research, Friedrich Schiller University, Loebdergraben 32, 07743 Jena, Germany

**Keywords:** transmission electron microscopy, dimensional stability, metal matrix composites, interferometry, cryogenics

## Abstract

In the present study, the thermal cycling stability of mirrors made of silicon particle reinforced aluminum compounds, containing an amount of 42 ± 2 wt.% silicon particles, is investigated with respect to thermal loading. The compound is processed by single-point diamond turning to optical mirrors that were subsequently thermally cycled in a temperature range between 40 °C to −60 °C and between 20 °C and −196 °C, respectively. The residual shape change of the optical surface was analyzed using Fizeau interferometry at room temperature. The change of shape deviation of the mirrors is compared with dilatometric studies of cylinders using the same temperature regime. Due to different coefficients of thermal expansion of the two constituents of the compound, thermal mismatch stresses in the ductile aluminum matrix and the brittle silicon particles are induced by the investigated thermal loads. The plasticity that occurs causes the formation of dislocations and stacking faults as substantiated by Transmission Electron Microscopy. It could be shown that the silicon particles lead to the cold working process of the reinforced aluminum matrix upon thermal cycling. By using interferometry, a higher dimensional stability of mirrors made of silicon particle reinforced aluminum due to thermal loads is demonstrated.

## 1. Introduction

Ultra-precise metal optics are particularly relevant for space applications and the scientific instrumentation of large telescopes are exposed to highly variable temperatures during their operational life. Shape deviations of <20 nm RMS (root mean square) at an area of 100 mm^2^ are commonly realized using a polishable electroless nickel plating on Al6061 (ISO AlMg1SiCu), a widely used substrate material for applications at ambient temperatures [1,2,3]. Unfortunately, metal optics of instruments used in large telescopes are often required to work under cryogenic conditions to avoid thermal radiation of the optical system itself [4]. Electroless nickel plated Al6061 shows a non-tolerable bimetallic bending under cryogenic conditions, due to the different coefficients of thermal expansion (CTE) of the substrate and the polishable layer [5]. In order to avoid those bending effects, a CTE matched aluminum substrate reinforced by silicon particles (42 ± 2 wt.% silicon) is used as substrate material for ultra-precise metal optics under cryogenic conditions [5,6,7].

Therefore, the dimensional stability in the nm-range of the substrate is crucial between ambient temperatures and −196 °C. Thermal cycling is known to result in stresses and plasticity in the aluminum matrix [8,9,10,11,12,13,14,15,16,17,18], which is caused by different coefficients of thermal expansion (CTE) of both phases, aluminum (CTE_Al_ = 22.9 × 10^−6^ K^−1^ at 20 °C) [19], and silicon (CTE_Si_ = 2.6 × 10^−6^ K^−1^ at 20 °C) [20]. Residual dimensional instabilities of silicon carbide (SiC) particle reinforced aluminum compounds were described in [21,22,23,24]. Residual length changes of particle reinforced aluminum materials in different temperature ranges were described in [21,22], whereas residual length changes of <1 × 10^−6^ (nm/mm) are required for mirrors [25]. It was previously shown that the residual deformation per cycle disappears almost entirely with an increasing number of thermal cycles [26,27]. Nevertheless, the influence of residual length changes on the shape deviation of mirrors made of silicon particle reinforced aluminum compounds was not described in detail yet.

In this work, a silicon particle reinforced aluminum compound containing an amount of 42 ± 2 wt.% silicon particles (Al-42Si) is investigated in terms of dimensional stability and (micro-)structural changes after thermal cycling. The purpose of this paper is (i) to investigate residual length changes of Al-Si42 caused by thermal cycling in different temperature regimes using dilatometry; (ii) to investigate the dimensional stability of the silicon particle reinforced aluminum compounds in different temperature regimes using Fizeau interferometry; and (iii) to characterize microstructural features of the compound before and after thermal cycling using Transmission Electron Microscopy (TEM), in order to correlate the findings to each residual changes.

## 2. Materials and Methods

### 2.1. Materials

The compound for the examination is a commercially available silicon particle reinforced aluminum compound, containing 42 ± 2 wt.% silicon. Al-42Si is produced via spray forming; subsequent hot isostatic pressing was carried out at temperatures above 500 °C; however, specific temperatures and annealing periods are proprietary to the manufacturers. The ‘as-received’ Al-42Si was processed in-house to nine metal optics (mirrors) with a diameter of 48 mm and a height of 15 mm by single-point diamond turning for interferometric analyses. For dilatometry, four cylindrical samples with a length of 25 mm and a diameter of 6 mm were prepared. Samples for electron microscopy, specifically Scanning Electron Microscopy (SEM), TEM, and Energy-dispersive X-ray spectroscopy (EDS), were prepared from materials of the same batches, as the mirrors were using the Focused Ion Beam (FIB) technique. A Helios NanoLab 600i system by FEI Co. (Hillsboro, OR, USA), operating with gallium ions under high-vacuum conditions, was used for enabling the lift-out method. For Al-42Si, one sample in the as-received condition (Al-42Si_AR_) and one sample after the thermal treatments (Al-42Si_TT_) were analyzed by TEM (see Table 1):

### 2.2. Methods

The microstructure was investigated via optical microscopy BX51 by Olympus, Tokyo, Japan) using ground and polished samples made of the same batch as the mirrors. For SEM and scanning TEM-EDS, the samples prepared by FIB were used. TEM analyses were carried out using a TEM Jem 3010 by JEOL Ltd. (Tokyo, Japan) (acceleration voltage: 300 kV), equipped with an EDS system.

The dimensional stability of Al-42Si was investigated using dilatometry (DIL 402 C by Netzsch Gerätebau GmbH, Selb, Germany) and Fizeau interferometry (DynaFiz^®^ by Zygo Corp., Middlefield, CT, USA). To compare residual length changes of Al-42Si cylinders to the residual changes of the shape deviation of metallic mirrors, three temperature regimes were applied to the cylinders: (i) annealing at 360 °C for up to twelve hours; (ii) environmental thermal cycling between −40 °C and 60 °C; (iii) cryogenic thermal cycling between −190 °C and 20 °C. To keep the atmosphere of the test chamber constant, it was purged with 30 mL/min helium. The residual length change ∆ε is analyzed observing the strain at 20 °C after several thermal cycles, using the following equation:∆ε = ε^(# + 1)^ − ε^#^(1)

The residual length change ∆ε is the difference of the sample’s length before (subscript number #) and after a thermal cycle (subscript number # + 1). To diminish systematic errors, the dilatometer was calibrated by measuring a Fused Silica sample under identical conditions.

Establishing a relation between the residual length change and the residual change of shape deviation of mirrors, the shape deviation of the optical surface of the manufactured mirrors at 20 °C was investigated using Fizeau interferometry. The shape deviation is defined as the difference of the real surface form to the ideal surface form of the mirror (in the case of mirrors, which are ideally prepared as being flat). Typical values are given as peak-to-valley values of the shape deviation or as RMS. We analyze and calculate the shape deviation using RMS values, because this value is less sensitive against contaminations on the optical surface. Using reference structures at the lateral surface of the mirrors, the optical surface was analyzed after thermal cycling in the same alignment. Thus, shape deviations of each sample are ultra-precisely traceable at an accuracy of ~1 nm RMS. Using this technique, plastic deformation and the according shape changes were investigated for the addressed thermal treatments. Three temperature regimes were applied to the samples: (i) annealing at 360 °C for up to twelve hours (annealing); (ii) environmental thermal cycling between −40 °C and 60 °C with cooling/heating rates <0.5 K min^−1^; (iii) cryogenic thermal cycling between −196 °C and 20 °C with cooling/heating rates <0.5 K min^−1^. The temperatures at thermal cycling were held at both the upper and lower limits for at least 1 h. 

For examining the stabilization effect of Al-42Si, the compounds were analyzed by TEM, regarding the occurrence of dislocations, twins, and/or stacking faults with respect to the thermal treatments. For distinguishing local strain fields of, e.g., dislocations from global strains, each sample was analyzed at two angles relative to the electron beam (90° and 80°). Crystallographic defects result in local strain fields that remain at a position, whereas global strain fields shift as a consequence of tilting the sample relative to the electron beam [28].

## 3. Results

### 3.1. Microstructure of Al-42Si

Figure 1 shows an optical micrograph of Al-42Si. The agglomerated silicon particles of Al-42Si are visible in the microstructure exhibit at a size of ≤50 µm.

The compound shows silicon particles, which coalesce in scattered elongated shapes due to the hot isostatic pressing procedure. After annealing, environmental, and cryogenic thermal cycling, optical microscopy did not reveal any changes to the microstructure of the compounds. The as-received condition (Al-42Si_AR_) and the thermally treated condition (Al-42Si_TT_) prepared by FIB were analyzed via SEM in secondary electron (SEM-SE) mode (Figure 2).

The distinct contrast visible in the microstructure allows for distinguishing aluminum and silicon phases, as confirmed by scanning TEM-EDS mappings (Figure 3).

Local quantitative measurements using EDS showed that the aluminum matrix and silicon particles are essentially pure phases. Additionally, the compound contains a small amount of Fe particles, which is a typical impurity [29].

### 3.2. Dimensional Stability upon Thermal Cycling

Dilatometric studies in a temperature range from 20 °C to 360 °C result in a residual length change of ∆ε ≤ 100 × 10^−6^ for Al-42Si. Fused Silica show a residual length change of ∆ε < 5 × 10^−6^ in the same temperature range, which represents the accuracy of the dilatometric technique. 

The strain vs. time diagram for Fused Silica and Al-42Si in a temperature range between −40 °C and 60 °C is shown in Figure 4, exemplarily.

After the first environmental cycle between −40 °C and 60 °C, Al-42Si shows a residual length change of <25 × 10^−6^. On the other hand, Fused Silica shows no significant residual length change after the first environmental cycle. Both materials, Al-42Si and Fused Silica, show residual length changes of ∆ε ≤ 5 × 10^−6^ from the second up to the twelfth environmental thermal cycle (Figure 5).

Obviously, after the first cycle onwards, the residual length change of both materials is close to or below the accuracy of the measurement method, because the measured residual length changes of Fused Silica and Al-42Si are similar. 

The first cryogenic thermal cycle of Al-42Si results in a residual length change of ∆ε < 100 × 10^−6^, compared with ∆ε < 5 × 10^−6^ for Fused Silica. An increasing number of cryogenic thermal cycles lead to a decreasing residual length change of Al-42Si. After the second cryogenic thermal cycle, the residual length change of Al-42Si is ∆ε < 5 × 10^−6^ (Figure 6).

After manufacturing the mirrors using single-point diamond turning, the shape deviation of the mirrors was analyzed by Fizeau interferometry (DynaFiz^®^ by Zygo Corp., Middlefield, CT, USA). at 20 °C and annealed at 360 °C. This thermal treatment led to a residual change of shape deviation of the mirrors at about 700 nm RMS, analyzed via a tactile 2.5D profilometer UA3P-5 by Panasonic Corp. (Osaka, Japan). For interferometric analysis, the resulting morphology and reflectivity were insufficient; therefore, all mirrors received a re-finishing by single-point diamond turning, enabling interferometric analysis after the intended thermal cycling. Contrary to the dilatometric studies in the same temperature regime, thermal cycling between −40 °C up to 60 °C produced a distinct dimensional change of the compound during the first cycle but, in subsequent cycles, a residual deformation did not occur (Figure 7).

It is important to note that after the first five thermal cycles, the change of shape deviation is normalized to one environmental thermal cycling step. From the second environmental thermal cycling step, Al-42Si exhibited dimensional stability within the accuracy of the interferometric technique of ≤1 nm RMS (Figure 8).

Increasing the thermal load of the previously cycled samples to a temperature range between −196 °C to 20 °C (cryogenic thermal cycling), resumed the dimensionally instable behavior for Al-42Si (Figure 9).

After the first cryogenic cycle, Al-42Si exhibited a change of shape deviation by 4 nm ± 2 nm RMS. From the third cycle on, Al-42Si exhibited dimensionally stable behavior during further cryogenic cycling, with the accuracy of the interferometric technique (Figure 10).

The investigated residual length changes determined by dilatometric studies of cylinders are consistent with the residual changes of shape deviation of the mirrors. An increasing number of thermal cycles leads to decreasing dimensional instability. 

### 3.3. TEM Analysis

In the following, representative TEM images of the aluminum matrix (Figure 11a) and the silicon particles (Figure 11b) are shown.

The dark stripes in Figure 11a are contrasts caused by dislocations contained in the aluminum matrix. The tiny black dots were identified as Ga-contamination (via scanning TEM-EDS) from the preparation procedure and were disregarded for the evaluation [30]. The silicon particles contained linear contrasts caused by stacking faults in the material (Figure 11b). Many of the contrasts are caused by large elastic strain fields and shifts during the tilting of the sample relative to the electron beam. Attaining a quantitative measure of the crystal defect density of each phase (dislocations, stacking faults) is thus ambiguous. For a general phenomenological analysis of the strengthening mechanism, however, TEM is the best option, because different crystal defects can be distinguished, and a qualitative comparison becomes possible. Table 2 shows a semi-quantitative classification of the investigated crystallographic defects in the aluminum matrix and silicon particles, respectively; both the as-received condition (Al-42Si_AR_) and the thermally treated condition (Al-42Si_TT_) are shown.

## 4. Discussion

Since details of the production route of the investigated silicon particle reinforced aluminum compounds are kept on a proprietary basis, one has to speculate regarding the causes of the microstructural features. Due to the eutectic temperature of the aluminum-silicon binary system and the melting point of aluminum [31], it can be reasonably assumed that the temperature at hot isostatic pressing is 500 °C ≤ T < 660 °C. This would lead to a partial re-melting of the aluminum matrix, and thus, a compact and dense material is more likely to be achieved [32]. The microstructure is a result of the high cooling rate during spray forming and coarsening (Ostwald ripening) during hot isostatic pressing. Silicon shows a low interdiffusion coefficient of 2.02 × 10^−4^ m^2^ s^−1^ in the aluminum phase [33]. Thus, it is assumed that the aluminum matrix contains a small but non-negligible silicon concentration of up to 1.5%. This increases the yield strength of the aluminum matrix compared to an essentially pure aluminum matrix. 

During cooling after hot isostatic pressing, thermal mismatch stresses due to different coefficients of thermal expansion of aluminum and silicon result in a significant number of dislocations in the aluminum matrix and stacking faults in the silicon particles caused by its coarsened microstructure. The resulting crystal defects, however, cannot be distinguished from those already present before cooling; their number might be reduced by dynamic recovery processes at the applied cooling rate from hot isostatic pressing temperatures.

Annealing at 360 °C leads to a residual length change of Al-42Si, resulting in a residual change of shape deviation of manufactured mirrors made of this material. 

Accordingly, the first thermal cycle between −40 °C and 60 °C causes a significant shape change for Al-42Si, with thermal mismatch stresses leading to further plastic deformation and increasing defect density. During environmental thermal cycling, plastic deformation of the ductile aluminum causes raises yield strength as a result of increasing defect density, leading to a work hardening effect at the investigated temperatures. This correlates with the declining change in shape deviation of the mirrors, subject to an increasing number of thermal cycles. A dimensionally unstable behavior during the cryogenic thermal cycling between −196 °C and 20 °C is caused by the higher thermal load. Higher thermal loads lead to higher thermal mismatch stresses [34]; therefore, work hardening occurs, similarly to thermal cycling between −40 °C and 60 °C. The thermally treated Al-42Si shows a significantly higher dislocation density, as detected in all considered locations of the aluminum phase. Stacking faults are observed solely in the silicon particles. Compared to the thermally untreated state of Al-Si42_AR_, a significantly higher number of crystal defects is observed. After sufficient strengthening of the compounds via work hardening, thermal mismatch stresses are compensated purely elastically, resulting in the dimensional stability of thermally treated mirrors made of Al-Si42_TT_. Fizeau interferometry shows a higher sensitivity to investigate suitable thermal treatments for the further development of dimensionally stable materials for ultra-precise mirrors.

## 5. Conclusions

Experimental results showed the dimensional stability dependence of the thermal procedures and microstructure of mirrors made of silicon particle reinforced aluminum compounds. Based on the analyses of the results, the following conclusions are drawn:

For ultra-precise mirrors, a material analysis of residual changes of shape deviation in the addressed temperature range is necessary. Thermal treatments at 360 °C and thermal cycling (−40 °C to 60 °C and −196 °C to 20 °C, respectively) of silicon particle reinforced aluminum materials lead to an increasing number of dislocations in the aluminum phase and to an increasing number of stacking faults in the silicon phase. Thermal cycling between −40 °C and 60 °C and −196 °C and 20 °C, respectively, lead to higher dimensional stability of silicon particle reinforced aluminum materials due to cold working of the ductile aluminum matrix caused by different coefficients of the thermal expansion of the two phases, aluminum and silicon.

## Figures and Tables

**Figure 1 materials-15-02998-f001:**
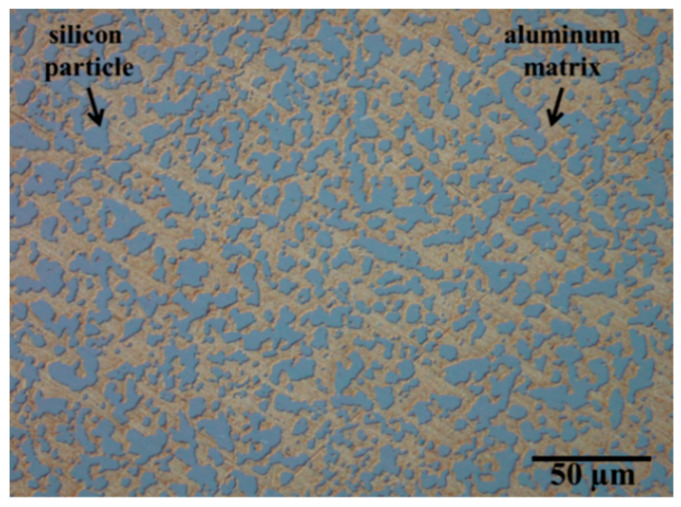
Optical micrograph of Al-42Si_AR_ (as-received).

**Figure 2 materials-15-02998-f002:**
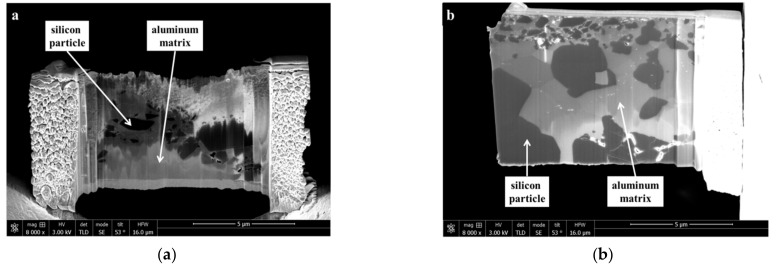
Scanning Electron Microscopy (SEM) secondary electron images of Al-42Si_AR_ (**a**) and Al-42Si_TT_ (thermally treated) (**b**). Samples were prepared by Focused Ion Beam (FIB).

**Figure 3 materials-15-02998-f003:**
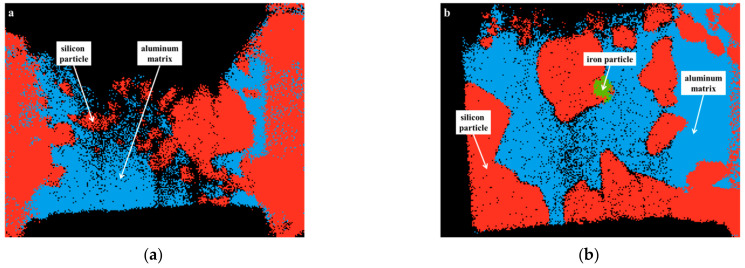
Transmission Electron Microscopy-Energy-dispersive X-ray spectroscopy (TEM-EDS) images of Al-42Si_AR_ (**a**) and Al-42Si_TT_ (**b**); aluminum: blue, silicon: red, iron: green.

**Figure 4 materials-15-02998-f004:**
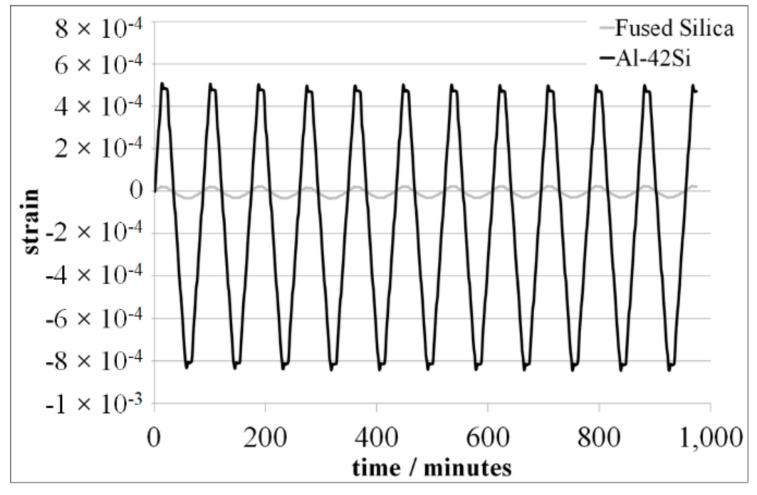
Strain behavior of Fused Silica and Al-42Si in a temperature range between −40 °C and 60 °C.

**Figure 5 materials-15-02998-f005:**
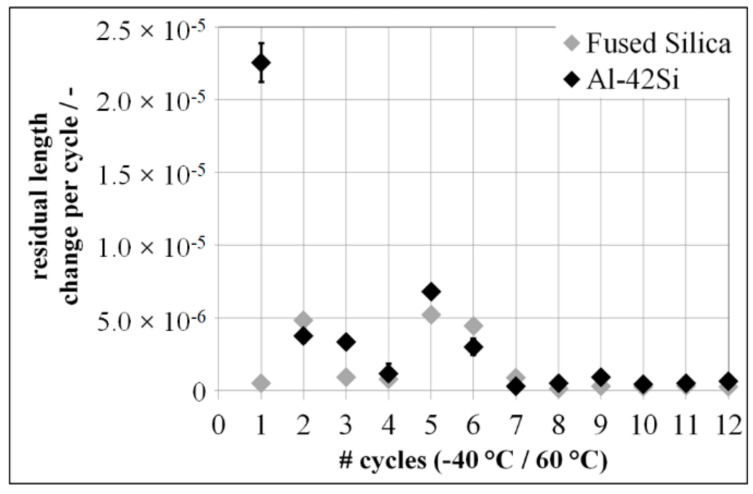
Residual length change of Fused Silica and Al-42Si in a temperature range between −40 °C and 60 °C.

**Figure 6 materials-15-02998-f006:**
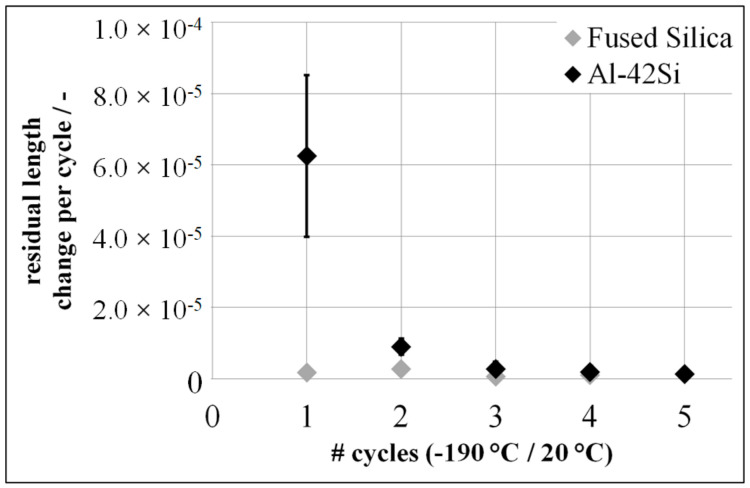
Residual length change of Fused Silica and Al-42Si in a temperature range between −190 °C and 20 °C.

**Figure 7 materials-15-02998-f007:**
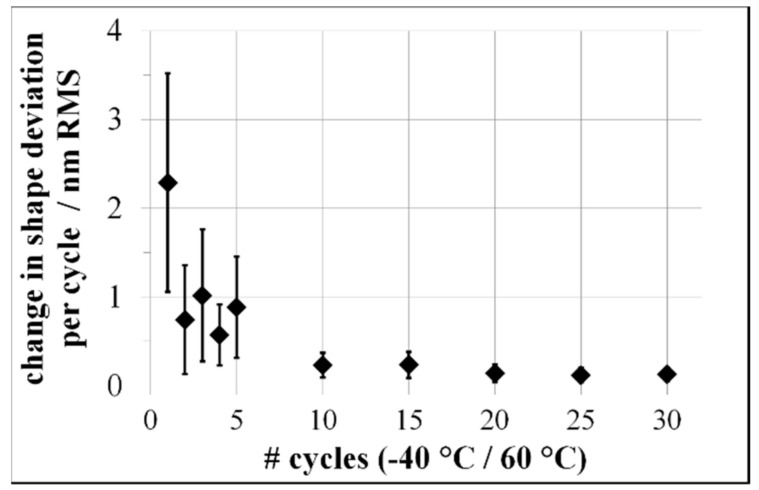
Change in shape deviation of mirrors made of Al-42Si with respect to numbers of environmental thermal cycles between −40 °C and 60 °C.

**Figure 8 materials-15-02998-f008:**
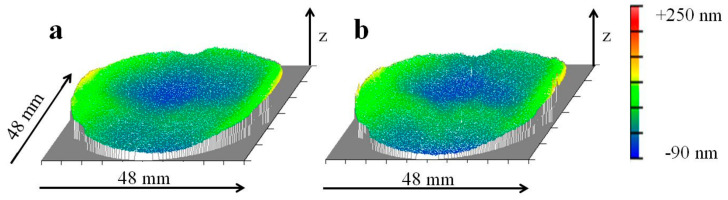
Residual change in shape deviation of the optical surface of a mirror made of Al-42Si, after second single-point diamond turning exhibiting 31 nm RMS (**a**); after 30 environmental thermal cycles exhibiting 34 nm RMS (**b**).

**Figure 9 materials-15-02998-f009:**
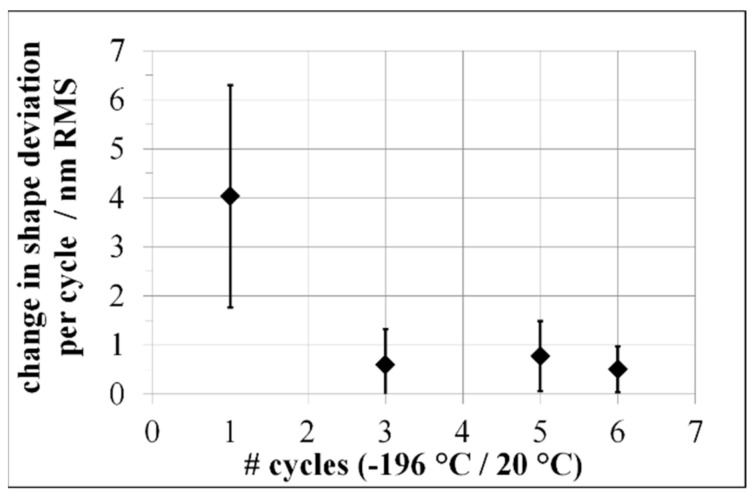
Change in shape deviation of mirrors made of silicon particle reinforced aluminum compounds after cryogenic thermal cycling.

**Figure 10 materials-15-02998-f010:**
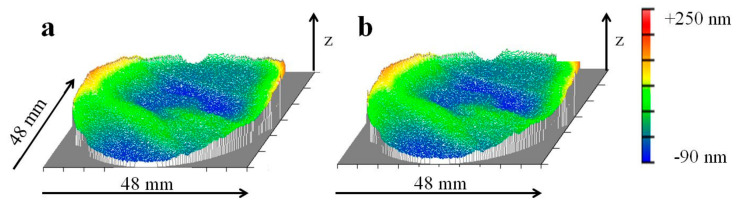
Dimensional stability of a mirror made of Al-42Si; after five cryogenic thermal cycles with 43 nm RMS (**a**); after six cryogenic thermal cycles with 43 nm RMS (**b**).

**Figure 11 materials-15-02998-f011:**
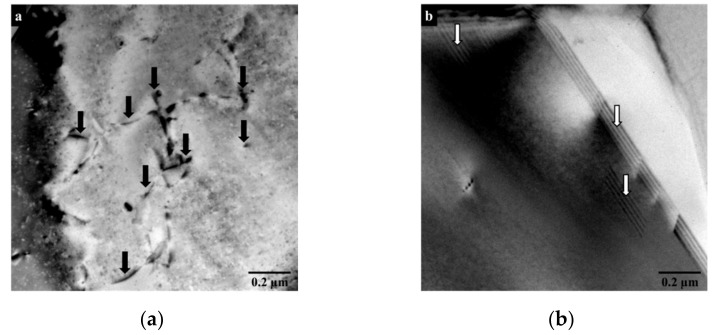
Dislocation contrasts in the aluminum matrix ((**a**): exemplarily marked by black arrows) and stacking faults in a silicon particle ((**b**): exemplarily marked by white arrows).

**Table 1 materials-15-02998-t001:** Thermal histories of as-received (Al-42Si_AR_) and thermally treated (Al-42Si_TT_) Al-42Si.

Thermal Treatment	Al-42Si_AR_	Al-42Si_TT_
annealing	-	360 °C
environmental thermal cycles	-	30 cycles between −40 °C and 60 °C
cryogenic thermal cycles	-	5 cycles between −196 °C and 20 °C

**Table 2 materials-15-02998-t002:** Semi-quantitative classification of the defect density in each phase of the samples in the as-received and thermally treated conditions, respectively.

	Al-42Si_AR_, Al Matrix	Al-42Si_AR_, Si Particle	Al-42Si_TT_, Al Matrix	Al-42Si_TT_, Si Particle
frequency of dislocations	-	-	++	+
frequency of stacking faults	--	-	--	++

Classifications: -- non, - marginal, + occasional, ++ frequently.

## Data Availability

Not applicable.

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
