# Peer review of "Dimensional Stability of Mirror Substrates Made of Silicon Particle Reinforced Aluminum"

_materials, 2022, doi:10.3390/ma15092998_

Round 1
Reviewer 1 Report
This paper is devoted to investigation of dimensional stability and microstructural changes of a silicon particle reinforced aluminum compound after thermal cycling. Experimental results showed the dimensional stability dependence of the thermal procedures and microstructure of mirrors made of silicon particle reinforced aluminum.
The article contains a number of interesting results that may be useful to a potential reader. The quality of the presentation of the material of the article, from the point of view of the reviewer, is good. The article may be published after minor editing.
There are several questions, comments and suggestions for the authors.
- In the text of the article there is a discrepancy between the results presented in fig. 4 and a description of this figure. The article test states: “In a temperature range between -40 °C and 60 °C, both materials Al-42Si and Fused Silica show residual length changes of ∆ε < 5 × 10-6 up to twelve environmental thermal cycles (Figure 4 )." However, in fig. 4 the amplitude of deformation changes for Al-42Si is approximately ∆ε ≈ 130 × 10-6. The authors need to resolve this contradiction.
- The presentation of the results, in the opinion of the reviewer, would have improved if the authors, in addition to Fig. 4 would represent dependence of Strain behavior of Fused Silica and Al-42Si in a temperature range between -196 °C and 20 °C and also from 20 °C to 360 °C.
- Authors need to inform readers how the confidence intervals in Fig. 5, 7 were determined: how the significance level (or confidence level) was set, how many measurements were made, what estimate (biased or unbiased) was used to calculate of the RMS.
- For a better understanding of the results presented, it seems appropriate to explain in more detail the content of the term "shape deviation".
- It is known that significant thermally induced stresses can occur as a result of significant differences in the coefficients of thermal expansion between the matrix and the strengthening particles. Thermal stresses, in turn, lead to an increasing number of dislocations and of twins / stacking faults. To the study of the influence of thermal stresses caused by the mismatch of CTE values on the formation of a dislocation structure have been devoted a significant number of publications. The authors are recommended to compare and analyze the obtained results with the data of other authors.
Author Response
- In the text of the article there is a discrepancy between the results presented in fig. 4 and a description of this figure. The article test states: “In a temperature range between -40 °C and 60 °C, both materials Al‑42Si and Fused Silica show residual length changes of ∆ε < 5 × 10-6 up to twelve environmental thermal cycles (Figure 4 )." However, in fig. 4 the amplitude of deformation changes for Al-42Si is approximately ∆ε ≈ 130 × 10-6. The authors need to resolve this contradiction.
Response by the authors:
The authors are grateful for this advice. We have added two diagrams (Figure 5 and Figure 6) to clarify details of the dilatometric results and the residual length changes.
- The presentation of the results, in the opinion of the reviewer, would have improved if the authors, in addition to Fig. 4 would represent dependence of Strain behavior of Fused Silica and Al-42Si in a temperature range between -196 °C and 20 °C and also from 20 °C to 360 °C.
Response by the authors:
We have added a residual length change vs. number of cycles diagram (-190 °C / 20 °C; Figure 6) to figure out the dilatometric results of the residual length changes more explicit. Regarding the thermal treatment at 360 °C, a diagram is not expedient, because each sample were measured just once.
- Authors need to inform readers how the confidence intervals in Fig. 5, 7 were determined: how the significance level (or confidence level) was set, how many measurements were made, what estimate (biased or unbiased) was used to calculate of the RMS.
Response by the authors:
The authors are grateful for this advice.The RMS values of Figure 7 and Figure 9 are described in chapter 2.2. The RMS value doesn’t represent the root mean square of the change of shape deviation per cycle results, but the shape deviation of each single optical surface. The error bars of Figure 7 and Figure 9 represent the standard deviations of the analyzed mirrors.
- For a better understanding of the results presented, it seems appropriate to explain in more detail the content of the term "shape deviation".
Response by the authors:
Many thanks for this suggestion. The shape deviation is described in chapter 2.2 in more detail.
- It is known that significant thermally induced stresses can occur as a result of significant differences in the coefficients of thermal expansion between the matrix and the strengthening particles. Thermal stresses, in turn, lead to an increasing number of dislocations and of twins / stacking faults. To the study of the influence of thermal stresses caused by the mismatch of CTE values on the formation of a dislocation structure have been devoted a significant number of publications. The authors are recommended to compare and analyze the obtained results with the data of other authors.
Response by the authors:
We have added some more actual publications. The shown results of the manuscript confirm the formerly described effects of thermal mismatch stresses in particle reinforced aluminum compounds (as described in chapter 1). Qualitatively, the mechanisms are consistent and can be traced back to dislocation formation in Al and stacking faults in Si. From the authors' point of view, a more in-depth quantitative comparison is currently not expedient. Basic mechanisms for the material system are accessed with high resolution using TEM. Consequently, the sample volume is small and thus limits the potential for quantification. A comparison of the dilatometer and interferometer results to the state of research seems currently not expedient, because a very small number of literature (e.g. [21] dilatometry on SiC particle reinforced Al; [23] interferometry on mirrors made out of SiC particle reinforced Al) is available and the complete thermal histories are not described, respectively.
Reviewer 2 Report
The submitted article discusses the dimensional stability of mirror substrates made of silicon par-ticle reinforced aluminum. The material is used in space applications and the reults could be of high importance. The manuscript is well written. However, the manuscript is a bit short especially when the reader knows about the very advanced techniques used in the experiments. The results must be improved and thus the discussion must be improved as well. Attached is an annotated PDF file contains the minor changes required on the manuscript.

Author Response
- It will be more informative if authors could tabulate this data.
Response by the authors:
The information are shown in Table 1.
- It is more professional to include the full name of the method when first mentioned and then use the abbreviation in the remaining text body.
Response by the authors:
The authors checked the abbreviations again and added the description of EDS (chapter 2.1).
- Define the parameters in equation 1.
Response by the authors:
Many thanks for the suggestion. We have added the definition of each parameter and the subscript numbers in chapter 2.2.
- The results are poorly discussed. Authors are encouraged to improve the results and discussion.
Response by the authors:
We have added some more actual publications. The shown results of the manuscript confirm the formerly described effects of thermal mismatch stresses in particle reinforced aluminum compounds (as described in chapter 1). Qualitatively, the mechanisms are consistent and can be traced back to dislocation formation in Al and stacking faults in Si. From the authors' point of view, a more in-depth quantitative comparison is currently not expedient. Basic mechanisms for the material system are accessed with high resolution using TEM. Consequently, the sample volume is small and thus limits the potential for quantification. A comparison of the dilatometer and interferometer results seems currently not expedient, because a very small number of literature (e.g. [21] dilatometry on SiC particle reinforced Al; [23] interferometry on mirrors made out of SiC particle reinforced Al) is available and the complete thermal histories are not described, respectively.
- Where did Fe come from? authors must explain that.
Response by the authors:
We are grateful for the suggestion. Iron is a typical impurity of silicon particle reinforced aluminum compounds. We have added a publication ([29]). By SEM-EDS, we found a very small number of iron particles. The FIB prepared lamella includes one of these very rare impurities.
- Authors are encouraged to improve the quality of this figure.
Response by the authors:
The quality of Figure 7 and Figure 9 is improved.
- Authors must perform the image stitching in a better way. Thus, these figures must be changed.
Response by the authors:
We have deleted the figure. The effect of tilting is solely described in chapter 3.3.
Reviewer 3 Report
- The sample names Al-42Si0 and Al-42Si1 should be revised. The subscript numbers are a little confusing.
- The sentence “Three temperature regimes were applied to the samples: (i) annealing at 360 °C for up to twelve hours (annealing); (ii) environmental thermal cycling between -40 °C and 60 °C with cooling/heating rates < 0.5 K min-1; (iii) cryogenic thermal cycling between -196 °C and 20 °C with cooling/heating rates < 0.5 K min-1.” appeared twice in the 2.2 section. It’s better to change a description for the second time.
- Please show the twins observed.
- The references should be updated as most of them are published in the 1990s or 2000s.
Author Response
- The sample names Al-42Si0 and Al-42Si1 should be revised. The subscript numbers are a little confusing.
Response by the authors:
We are grateful for this suggestion. The subscript numbers were changed Al-42Si0 à Al-42SiAR (as-received); Al-42Si1 à Al-42SiTT (thermally treated) and defined in chapter 2.1 (Table 1).
- The sentence “Three temperature regimes were applied to the samples: (i) annealing at 360 °C for up to twelve hours (annealing); (ii) environmental thermal cycling between -40 °C and 60 °C with cooling/heating rates < 0.5 K min-1; (iii) cryogenic thermal cycling between -196 °C and 20 °C with cooling/heating rates < 0.5 K min-1.” appeared twice in the 2.2 section. It’s better to change a description for the second time.
Response by the authors:
We did not change the formulation. From the authors' point of view, it seems more clear to describe the thermal treatments for the mirrors and the interferometer analyses as proposed. The atmospheres (helium purged at dilatometry vs. ambient at thermal treatments of mirrors) and the minimum temperature ‑190 °C at dilatometry vs. ‑196 °C at thermal treatments of mirrors) are less different.
- Please show the twins observed.
Response by the authors:
In comparison to stacking faults, very few twins were observed in Si particles. We cannot completely rule out that twinning plays a minor role, however, it is clear that the formation of stacking faults dominates. We have therefore adapted the text, now omitting explicitly mentioning twins.
- The references should be updated as most of them are published in the 1990s or 2000s.
Response by the authors:
We have added four publications ([17], [18], [24], and [34]) published between 2018 and 2021.